# Real-World Experience with the Available Outpatient COVID-19 THErapies in Patients with canceR (CO.THER)

**DOI:** 10.3390/cancers17060999

**Published:** 2025-03-17

**Authors:** Angioletta Lasagna, Giulia Gambini, Catherine Klersy, Simone Figini, Sofia Marino, Paolo Sacchi, Paolo Pedrazzoli

**Affiliations:** 1Medical Oncology Unit, Fondazione IRCCS Policlinico San Matteo, 27100 Pavia, Italy; simone.figini01@universitadipavia.it (S.F.);; 2Biostatistics and Clinical Trial Center, Fondazione IRCCS Policlinico San Matteo, 27100 Pavia, Italy; gi.gambini@smatteo.pv.it (G.G.); c.klersy@smatteo.pv.it (C.K.); 3Division of Infectious Diseases I, Fondazione IRCCS Policlinico San Matteo, 27100 Pavia, Italy; p.sacchi@smatteo.pv.it; 4Department of Internal Medicine and Medical Therapy, University of Pavia, 27100 Pavia, Italy

**Keywords:** COVID-19, nirmatrelvir–ritonavir, long COVID, cancer, remdesivir

## Abstract

This article regards the role of early anti-SARS-CoV-2 therapies in cancer patients undergoing active therapy. From our study, these therapies appear effective and safe in preventing the hospitalization of cancer patients, who are known to be more fragile and at risk of severe complications from COVID-19. In addition, we evaluated the impact of these therapies in reducing symptoms of long COVID. The overlap between the symptoms related to the oncological disease/oncological treatment and the symptoms of long COVID is one of the main future challenges oncologists will have to manage.

## 1. Introduction

Cancer represents an important risk factor for acquiring severe acute respiratory syndrome by Coronavirus-2 (SARS-CoV-2) and subsequent hospitalization and poor outcomes [1,2,3]. COVID-19 vaccines substantially changed the course of the pandemic. Thanks to a mathematical model of COVID-19 transmission and vaccination, vaccines played an essential role in saving lives during the first year of their introduction [4]. However, the immunogenicity and efficacy of vaccines may be variable in patients with cancer and may vary over time [5,6]. In response to the pandemic, several early antiviral therapies have been licensed to treat COVID-19, with varying degrees of efficacy. The aim is to start antiviral treatment in the early stages of infection, ideally within the first few days after the onset of symptoms, to prevent progression to severe disease. Remdesivir [7], Sotrovimab [8], Molnupinavir [9], and Nirmatrelvir–Ritonavir [10] are the main medications. To date, the utility of these agents in patients with cancer has not yet been clearly demonstrated. Remdesivir is a SARS-CoV-2 nucleotide analog RNA polymerase inhibitor and is able to prevent the replication of the virus [11]. The use of Remdesivir in cancer patients requires careful evaluation due to the risk of potential drug interactions and of renal toxicity because it is excreted through the kidneys [12]. Early antiviral intervention in frail populations such as patients with cancer was evaluated in a retrospective study in England. The authors included patients treated with sotrovimab, Nirmatrelvir–Ritonavir, and Molnupiravir. They reported a particularly low COVID-19-related hospitalization rate for patients receiving Sotrovimab (0.7%) and Nirmatrelvir/Ritonavir (0.3–1.2%) [13]. Generally, it should be emphasized that these antiviral treatments were not initially tested in cancer patients, particularly in immunocompromised patients. Therefore, real-life studies to evaluate the safety and efficacy of these treatments in cancer patients are of great value. Treatment regimens should be tailored to the patient’s cancer treatment history, general health, and specific risk factors for severe COVID-19.

In addition to the severe clinical pictures that can occur in immunocompromised patients in the acute phase of COVID-19, such as Acute Respiratory Distress Syndrome (ARDS) [14], symptoms after this phase may persist in the form of long COVID. It is a multisystemic condition that can persist over several months after acute infection [15]. For patients with cancer, the impact of long COVID can be particularly disturbing due to their already compromised health condition and the additional burden of cancer treatment and its side effects. For example, one of the typical symptoms of long COVID is chronic fatigue, which can be even more debilitating in patients with cancer. Chemotherapy and other oncological treatments already induce fatigue, so the added burden of long COVID symptoms can affect the quality of life by impairing daily activities. The protective effect of early therapies on long COVID outcomes is still debated, with contradictory results on both real effectiveness and durability [16,17].

We conducted the CO.THER study (COVID-19 THErapies in patients with canceR) to address this knowledge gap, focusing on the safety of these therapies in combination with anticancer therapies, and the clinical outcome of these patients.

## 2. Materials and Methods

### 2.1. Study Design and Setting

This is an ambispective single-center cohort study. All the patients undergoing active oncological treatment at the Medical Oncology Unit of Fondazione “IRCCS Policlinico San Matteo di Pavia” with a symptomatic COVID-19 infection from January 2022 to January 2023 (Omicron-dominated period) and treated with an early anti-SARS-CoV-2 therapy as indicated by the specialist in infectious diseases were enrolled. Nirmatrelvir–Ritonavir was prescribed per os at a dose of 300/100 mg every 12 h for 5 days, Molnupiravir was prescribed per os at a dose of 200 mg every 12 h for 5 days, and Remdesivir was administered intravenously for 3 days (200 mg on day 1, and then 100 mg from day 2 to day 3). We conducted the study according to the Strengthening the Reporting of Observational Studies in Epidemiology (STROBE) Statement for reporting observational studies [18]. The local Ethics Committee (Comitato Etico Area Pavia) and Institutional Review Board (P-0039959/22) approved the study (CO-THER). All the subjects signed the informed consent.

This paper reported the conclusive data of all the 131 enrolled patients.

The primary endpoint evaluated the rate of hospitalization for COVID-19 disease within 14 days in cancer patients using anti-SARS-CoV-2 early therapies. The secondary objectives were as follows: (i) evaluation of the mortality rate for COVID-19 disease; (ii) evaluation of the time to COVID-19 symptom resolution; (iii) evaluation of the time to nasal swab viral clearance; (iv) evaluation of the time of the re-start of the oncological treatment; (v) evaluation of the incidence of drug-related adverse events; and (vi) evaluation of the incidence of long COVID symptoms according to WHO case definition.

Our exploratory endpoint was to evaluate whether there was a difference between re-infected subjects and those who had been infected only once compared to all the endpoints mentioned above.

### 2.2. Data Collection

We collected clinical and oncological data from the hospital’s electronic patient records. Data were collected at T0 = start of COVID-19 therapy; T1 = seven days after T0; T2 = 2 weeks after T0; T3 = one month after T0; T4 = +3 months after T0; T5 = +6 months after T0; T6 = +12 months after T0.

Symptoms of long COVID queries included fatigue, dyspnea, cough, cardiovascular symptoms, gastrointestinal (GI) disorders (such as nausea, vomiting, and diarrhea), neuro-psychological symptoms (such as brain fog, and memory and concentration troubles), headache, altered taste and smell, rheumatologic symptoms, dermatologic symptoms, and other.

The inclusion criteria were as follows: (i) patients aged 18 and older, regardless of gender; (ii) oncological therapies; (iii) patients who received one of the early anti-SARS-CoV-2 therapies (Sotrovimab, Molnupiravir, Remdesivir, and Nirmatrelvir–Ritonavir) according to the infectious disease specialist consulting for mild-to-moderate COVID-19; (iv) signing of informed consent. The exclusion criteria included experiencing symptoms for more than 5 days, hospitalization due to COVID-19, and the absence of severe liver disease or renal failure. Indeed, these therapies are not recommended if the onset of symptoms is longer than 5 days [19]. The exclusion of patients already hospitalized was due to not wanting to consider patients with more severe manifestations of the disease. Finally, severe liver disease or renal failure can significantly impact the metabolization of anti-viral therapies. The exclusion criteria mentioned were chosen to make the examined population more homogeneous. In fact, we excluded subjects who might not have benefited from the treatment and those whose underlying conditions might have interfered with the results of the study.

The variables analyzed included demographic characteristics, COVID-19-related symptoms, comorbidities, dates of symptom onset and drug administration, and anti-SARS-CoV-2 vaccination. The diagnosis of COVID-19 was performed with real-time reverse-transcription polymerase chain reaction (RT-PCR) testing of nasopharyngeal swabs using the Cobas^®^8800 System/cobas SARS-CoV-2 (Roche Diagnostics, Basel, Switzerland). Nasopharyngeal swabs were repeated when symptoms disappeared and until they became negative.

### 2.3. Statistical Analysis

Analyses were performed using the Stata software (release 18, StataCorp., College Station, TX, USA). A 2-sided *p*-value < 0.05 was considered as statistically significant. Categorical variables were described as counts, percentages, continuous variables, medians, and quartiles (IQR). The proportion of hospitalizations within 14 days (primary endpoint) was computed together with its exact binomial 95% confidence interval (95%CI). Rates per 100-person year were computed for time-to-event outcomes and the median (event-free) survival was computed, with its IQR, when reached. The Kaplan–Meier cumulative (event-free) survival was computed with 95%CI and plotted. Comparisons were performed using the log-rank test, and hazard ratios (HRs) and 95% CIs were derived from a Cox model.

## 3. Results

### 3.1. Characteristics of the Study Population

One hundred and thirty-one patients’ records (53 males [40.5%], 78 females, [59.5%]; median age 62.45 years, interquartile range [IQR] 56–71) were enrolled. Seventy-seven patients (58.8%) had ≥1 comorbidity, and the majority (28 patients, 36.4%) had chronic obstructive pulmonary disease (COPD). All the clinical characteristics are shown in Table 1.

Forty-three patients (32.8%) had breast cancer, thirty-one (23.7%) had non-small cell lung cancer (NSCLC), sixteen (12.2%) had colorectal cancer, seven (5.3%) had pancreatic cancer, six (4.6%) had melanoma, five (3.8%) had kidney cancer, and four (3.1%) had head and neck cancer. The remaining seventeen patients had urothelial/bladder cancer (3 patients, 2.3%), small cell lung cancer (SCLC, 3 patients, 2.3%), gastric cancer (3 patients, 2.3%;), hepatocellular cancer (HCC, 3 patients, 2.3%), or other types of cancer (7 patients, 5.3%). The majority of patients had stage IV cancer (90 patients, 68.7%). The most common treatment was chemotherapy (69 patients, 52.7%; Table 2).

### 3.2. SARS-CoV-2 Vaccination, Infection, and Early Therapies

Thirty-seven patients (28.2%) had experienced a previous SARS-CoV-2 infection (the first wave) before being enrolled in this study. The average number of cycles received at the time of enrollment in this study was 10 (IQR 2–12). One hundred and nine patients (83.2%) had received three doses of COVID-19 vaccine, while only eighteen patients (13.7%) had received a fourth dose of COVID-19 vaccine at the time of data collection.

All the enrolled patients had a symptomatic COVID-19 infection. In particular, one hundred and twenty-one patients (92.4%) had rhinorrhea, eighty-seven (66.4%) had a cough, eighty-six (65.6%) had a fever, forty-six (35.1%) and thirty-eight (29%) had fatigue and sore throat, respectively. Twenty-five patients (19.1%) reported headache, while fourteen (10.7%) and eleven (8.4%) patients reported arthralgia and myalgia, respectively. Only two patients (1.5%) had gastrointestinal symptoms with nausea and vomiting at the time of COVID-19 infection.

One hundred and four patients (79.4%) received Nirmatrelvir–Ritonavir, fifteen patients (11.4%) received Remdesivir, while nine (6.9%) and three (2.3%) patients received Molnupinavir and Sotrovimab, respectively.

### 3.3. Primary Endpoint: Hospitalization Within 14 Days

As shown by the Kaplan–Meier hospitalization free estimate, only three patients (2.1%) were hospitalized for a COVID-19-related cause within 14 days of starting early treatment (95%CI 0.5–6.6%) (Figure 1). The cumulative survival probability beyond 12 months in hospitalization-free patients was 98% (95%CI 93–99%).

### 3.4. Secondary Endpoints: SARS-CoV-2 Infection

During the follow-up, twenty-four patients (18.5%) died, of which only one patient (4.2%) died of COVID-19 disease. The mortality rate per 100 person months was 15.7% (95%CI 10.5–23.4%). Five patients had experienced a previous SARS-CoV-2 infection, while the other nineteen did not. We compared the probability of survival in those patients who had experienced a previous SARS-CoV-2 infection versus those who had not. We found a probability of survival of 86% at 12 months in the group with previous infection (95%CI 70–94) and 78% (95%CI 68–86) in the group without previous infection (HR 0.61, SD 0.31, 95%CI 0.23–1.63; *p* = 0.32) (Figure 2). This association was not confounded by previous vaccination (HR 0.59, 95%CI 0.22–1.58).

The median time to COVID-19 symptom resolution was six days (IQR 5–9 days), while the median time to nasal swab viral clearance was seven days (IQR 6–11 days). Only one patient did not have a resolution of symptoms and viral clearance of the follow-up. The median time to the re-start of the oncological treatment was fourteen days (IQR 12–16 days). Only four patients did not restart the oncologic treatment during the observed follow-up. No drug-related adverse events were observed during the follow-up. Twelve patients (9.2%) reported another SARS-CoV-2 infection during the follow-up and they were all retreated with Nirmatrelvir–Ritonavir. The cumulative reinfection-free survival was 90% at 12 months (95%CI 83–95%).

### 3.5. Secondary Endpoints: Long COVID

Further, 15 patients of the 123 evaluable at 3 months (median age 51 years, IQR 40–68) reported long COVID symptoms (12.2%, 95%CI 7.0–19.3). They were all female. Thirteen patients (86.6%) had breast cancer, one patient (6.7%) had NSCLC, and one patient (6.7%) had colorectal cancer. Four patients (26.7%) had metastatic disease. With regard to cancer treatment, most patients were receiving chemotherapy or hormone therapy. Nine patients (60%) reported fatigue, three patients (20%) had headaches, and twelve patients (80%) reported neuropsychological symptoms (brain fog, insomnia, and memory and concentration troubles).

## 4. Discussion

In our previously published papers, we reported preliminary data about the effectiveness of the anti-SARS-CoV-2 therapies in the first six months following the AIFA prescribability and the incidence of long COVID after three months of follow-up in part of this cohort [20,21]. In this paper, we analyzed the impact of early anti-SARS-CoV-2 therapies on the COVID-19-related 14-day hospitalization rate in cancer patients. Early antiviral therapy is most effective when ideally administered within 48 h of symptom onset, but for our population, identifying COVID-19 early can be challenging due to overlapping symptoms with cancer and cancer treatments.

In this cohort of high-risk subjects, we report a low rate of hospitalizations (2.3%) and no drug-related adverse events. This study was conducted during the circulation of the Omicron variant, which has a reduced risk of hospitalization and death [22]. Despite this observation, cancer patients are at higher risk of hospitalization, even if infected with the Omicron variant [23]. Some recent papers have focused on the usefulness of these drugs in immunocompromised patients including those with solid tumors. EPIC-SR (Evaluation of Protease Inhibition for COVID-19 in Standard-Risk Patients) trial reported the clinical benefit of Nirmatrelvir–Ritonavir in older patients at a higher risk of COVID-19 progression [24]. In a study with a sample size of just sixty-seven cancer patients, Guermazi and colleagues [25] demonstrated the efficacy of oral antivirals for COVID-19, similarly to what we highlighted.

A limit to the use of early antiviral therapy is the occurrence of drug–drug interaction, above all when Nirmaltrevir–Ritornavir was employed. Indeed, Ritornavir acts as an inhibitor of the Cyp450 3a subunit, limiting its use in patients undergoing polypharmacy [26]. Therefore, our data seem to reassure us of the efficacy of early anti-SARS-CoV-2 therapies, even in cancer patients undergoing oncological active treatments. These results were also demonstrated by a recent retrospective cross-sectional study. Zaharuddin and colleagues highlighted the safety and tolerability of Nirmatrelvir–Ritonavir with no severe adverse events [27]. Our experience confirms the safety of this drug if the pharmacological history is carefully collected and patients are managed by a multidisciplinary team including oncologists, pharmacists, and infectious disease specialists.

Although the cumulative survival was higher in patients with repeated COVID-19 infections, we do not have the power to claim this difference as statistically significant. Numerous studies evaluated the role of vaccination in patients with cancer. In a meta-analysis, Hua and colleagues demonstrated significantly lower seroconversion rates after COVID-19 vaccination in patients with solid tumors [28]. The hybrid immunity (previous infection + vaccination) might enhance the immune response and thus survival [29], though these data need further confirmation in large cohorts. Jorda and colleagues demonstrated a significantly higher benefit of early antiviral therapies in patients with no immunization or with expired immunization, defined by the authors as “SARS-CoV-2 vaccines or disease > 6 months ago” [30]. The role of vaccination in preventing long COVID is still much debated. Ceban and colleagues described in a meta-analysis that vaccination against SARS-CoV-2 is associated with a lower incidence of long COVID compared to unvaccinated subjects, but the magnitude of the protective effect varied between studies and did not take into account any early anti-viral therapy [31]. This finding should be evaluated by prospective studies to assess the real combined role of infection, vaccines, and early antiviral treatment.

Finally, we estimated the incidence of long COVID in this cohort of patients. All patients with long COVID were females. This evidence is in line with other papers in the literature [32,33]. There is a striking similarity between some symptoms of long COVID and the symptoms related to oncological disease/oncological treatment. This may have created difficulties in the differential diagnosis. Cognitive impairment, depression, and (mental) fatigue are not only the most frequent neuropsychiatric symptoms of long COVID [34], but patients with cancer typically also report them during their oncological treatments [35]. We demonstrated that the incidence of these symptoms is very low and does not affect patient management. In a large target trial emulation framework, Wang and colleagues investigated the protective role of Nirmatrelvir–Ritonavir against long COVID. In this preprint, they reported a long COVID risk reduction with Nirmatrelvir–Ritonavir in high-risk patients such as patients with cancer [36].

This study has several strengths. We enrolled only patients with solid tumors, to make the sample size as homogeneous as possible. We focused on a specific population that has not been specifically analyzed in previous studies on COVID-19. We provided evidence of the efficacy and safety of early antiviral therapies in this patient population to improve their clinical management. This study has some limitations. It is a single-center cohort study and thus has a limited sample size of 131, which does not allow for narrow confidence intervals. However, this single-center study guarantees a higher homogeneity of behaviors. We do not have a control arm, which would have been unethical at the time. We were not able to perform daily nasopharyngeal swabs to promptly assess negative test results, but we acknowledge that the design of our study was not intended to evaluate viral shedding. Although viral load and time to negative test results are important, the primary objective of our study was to evaluate the efficacy of antiviral therapies. Daily testing was not essential to achieve this goal. Finally, larger collaborative multicenter cohort studies will be able to confirm these initial observations.

## 5. Conclusions

These real-life data can help us understand how to focus all our efforts on cancer patients. The management of COVID-19 in patients with cancer requires a delicate balance of early antiviral intervention, cancer treatment protocols, and individualized patient care. With these antiviral therapies, it is possible to improve outcomes in this vulnerable population, but a careful consideration of drug interactions and the proper timing of treatment is essential. Continued research and clinical trials are critical to optimize the care of cancer patients during the COVID-19 pandemic. A careful examination of long COVID symptoms may allow for a personalized approach to improving the quality of life of cancer patients by enabling the continuation of treatment.

## Figures and Tables

**Figure 1 cancers-17-00999-f001:**
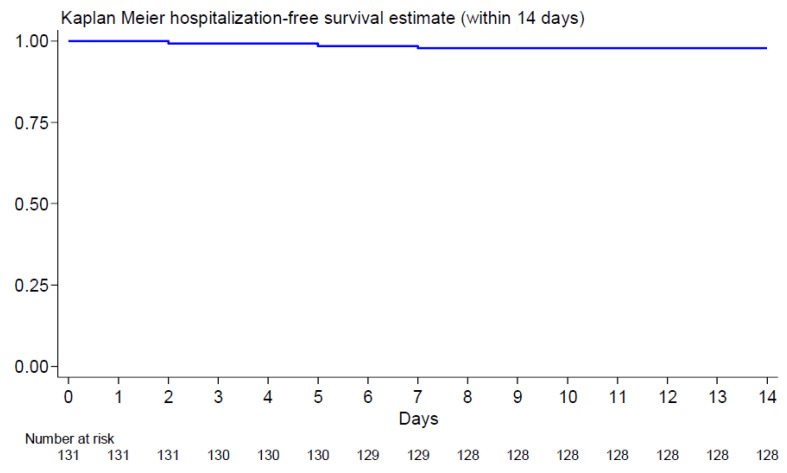
Kaplan–Meier hospitalization-free survival estimate: In this figure, we observed hospitalization only for three patients (95%CI 0.5–6.6%).

**Figure 2 cancers-17-00999-f002:**
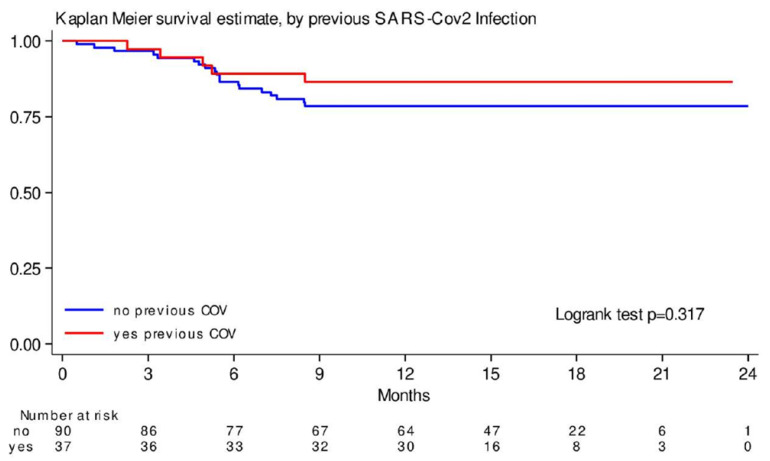
Kaplan–Meier survival estimate, by previous SARS-CoV-2 Infection: This figure points out the probability of survival at 12 months in the group with previous infection (red line) and in the group without previous infection (blue line) (HR 0.61, SD 0.31, 95%CI 0.23–1.63; *p* = 0.32).

**Table 1 cancers-17-00999-t001:** Patients’ general clinical characteristics.

%	Num Patients	Patients’ Characteristics
59.5/40.5	78/53	Female/Male
34.3/65.7	45/86	Smoke (yes/no)
44.4/55.6	20/25	Smokers (former/current)
41.2	54	No comorbidities
58.8	77	≥1 comorbidity
29.9	23	Diabetes mellitus type 2
29.9	23	Cardiovascular diseases
5.2	4	Autoimmune diseases
36.4	28	COPD ^1^
12.9	10	Neurological diseases (like ansia, depression, etc.)
3.9	3	Dermatologic diseases
2.7	2	Cerebrovascular diseases

^1^ Legend: COPD: chronic obstructive pulmonary disease.

**Table 2 cancers-17-00999-t002:** Oncological patient characteristics.

%	Num Patients	Type of Tumor
32.8	43	Breast cancer
23.7	31	NSCLC
12.2	16	Colon–rectal cancer
5.3	7	Pancreatic cancer
4.6	6	Melanoma
3.8	5	Kidney cancer
3.1	4	Head and neck cancer
3.2	3	Urothelial/bladder cancer
3.2	3	HCC
3.2	3	Gastric cancer
3.2	3	SCLC
5.3	7	Others
		Tumor Stage
31.3	41	Stage I/II/III
68.7	90	Stage IV
		Type of oncological treatment
52.7	69	Chemotherapy
25.9	34	ICIs
15.3	20	Hormone therapy
19.8	26	Targeted therapy
		Type of ICIs
82.4	28	Anti PD-1
14.7	5	Anti PD-L1
2.9	1	Anti CTLA4

Legend: NSCLC: non-small cell lung cancer; SCLC: small cell lung cancer; HCC: hepatocellular carcinoma; ICI: immune checkpoint inhibitors.

## Data Availability

All the data supporting the findings of this study can be found within the article.

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
