# Peer review of "Real-World Experience with the Available Outpatient COVID-19 THErapies in Patients with canceR (CO.THER)"

_cancers, 2025, doi:10.3390/cancers17060999_

Round 1
Reviewer 1 Report
Comments and Suggestions for Authors
Review for cancers-3507206
In this original article entitled “Real-world experience with the available outpatient COvid-19 THErapies in patients with canceR (CO.THER)”, the authors (Lasagna et al.) assessed the safety of combined anticancer and covid-19 therapies (COvid-19 THErapies in patients with cancer; CO.THER) and studied the clinical outcome of these included patients. The authors reported a low rate of hospitalizations and reassuring data of safety in cancer patients, who are known to be fragile and at risk of severe complications from COVID-19
The manuscript is acceptable but requires some improvements. Hereafter some comments revealed after reviewing its current version.
- The major lack of the study is the low number of patients, who are also belonging to a single-center cohort study.
- Regarding the exclusion criteria, the authors mentioned that: “The exclusion criteria were …, and absence of severe liver disease or renal failure.”. It is recommended to indicate also the absence of any associated cancerous diseases rather than severe liver disease or renal failure. Have the patients with severe liver disease or renal failure been included? More clarification(s) is/are needed
- Definitely, the readers would like to the distribution of the cancer stages of the studied patients. Why do stages I, II and III have been combined? It would be better to add such data as this may mediate the results.
- It is recommended to use covid-19 instead of covid only in the whole manuscript because sometimes this is confusing such as for the COVID related-symptoms but not with COVID-19 Symptoms…
- The sentences “Cancer represents an important … and poor outcomes.” and “Symptoms after the acute phase … as long COVID” can be supported by the following relevant and recent references; doi: 10.1080/07391102.2020.1803139.
- English language is overall acceptable just minor editing and checking are required just like the sentence “Symptoms after the acute … as long COVID.”
Author Response
The major lack of the study is the low number of patients, who are also belonging to a single-center cohort study.
Author response: We recognize that the monocentric design is a limitation of the work and therefore emphasized this aspect in the discussion. We have now extended this paragraph: “It is a single-center cohort study and thus with a limited sample size of 131 that does not allow for narrow confidence intervals. However, this one center study would guarantee a higher homogeneity of behaviors. We do not have a control arm, which would have been unethical at the time.”
Regarding the exclusion criteria, the authors mentioned that: “The exclusion criteria were …, and absence of severe liver disease or renal failure.”. It is recommended to indicate also the absence of any associated cancerous diseases rather than severe liver disease or renal failure. Have the patients with severe liver disease or renal failure been included? More clarification(s) is/are needed
Author response: The exclusion criteria mentioned were chosen to make the population examined more homogeneous. In fact, we excluded subjects who might not have benefited from the treatment or those whose underlying conditions might have interfered with the results of the study, to ensure the relevance of the results.
Definitely, the readers would like to the distribution of the cancer stages of the studied patients. Why do stages I, II and III have been combined? It would be better to add such data as this may mediate the results.
Author response: Given the small sample size, we have chosen to divide the examined population into two groups, similar to other studies.
It is recommended to use covid-19 instead of covid only in the whole manuscript because sometimes this is confusing such as for the COVID related-symptoms but not with COVID-19 Symptoms…
Author response: For the entire manuscript we used COVID-19 when referring to the clinical manifestation from SARS-CoV-2 and long COVID when referring to ‘post-acute sequelae of SARS-CoV-2 infection’, in accordance with what other authors have reported (e.g. PMID: 35874958)
The sentences “Cancer represents an important … and poor outcomes.” and “Symptoms after the acute phase … as long COVID” can be supported by the following relevant and recent references; doi: 10.1080/07391102.2020.1803139.
Author response: We improved the introduction and added the reference
English language is overall acceptable just minor editing and checking are required just like the sentence “Symptoms after the acute … as long COVID.”
Author response: English has been revised in order to correct stylistic errors.
Reviewer 2 Report
Comments and Suggestions for Authors
The current study evaluates the effectiveness and safety of early anti-SARS-CoV-2 therapies in cancer patients undergoing active treatment. Conducted as an ambispective single-center cohort study, it assesses hospitalization rates, mortality, symptom resolution, and long COVID outcomes. Among 131 patients, only 2.1% required hospitalization within 14 days, with a 12-month hospitalization-free survival rate of 98%. Long COVID symptoms were reported in 12.2% of patients, predominantly affecting women. The study suggests that early antiviral treatment is safe and effective for high-risk cancer patients, though further multicenter research is needed to confirm these findings.
- In the abstract, further details are required in the methods.
- The keywords should be indicative of the manuscript content. (e.g. brain fog and outcome should be removed).
- The statement of the exclusion criteria is confusing. Please revise and rephrase.
- A comprehensive conclusion is required.
Author Response
In the abstract, further details are required in the methods
Author response: We added more information in the methods
The keywords should be indicative of the manuscript content. (e.g. brain fog and outcome should be removed).
Author response: We removed brain fog and outcome
The statement of the exclusion criteria is confusing. Please revise and rephrase.
Author response: We have further justified the exclusion criteria in the appropriate section
A comprehensive conclusion is required.
Author response: We improved the discussion and the conclusion sections.
Reviewer 3 Report
Comments and Suggestions for Authors
Reviewer Comments
This topic is of some interest and the manuscript is easy to follow.
However, I have the following comments:
1.The manuscript provides real-world data on long COVID outcomes in cancer patients, a poorly understood area and examines drug safety in combination with active cancer therapies, addressing concerns. However, there are still some shortcomings, such as, the single-center, ambispective design and lack of a control group reduce the novelty compared to randomized trials.
- There are some suggestions for improvement regarding the research methodology.Control Group: Include untreated or placebo-controlled patients to isolate the effect of antiviral therapies. Multicenter Collaboration: Expand recruitment to improve generalizability and reduce selection bias. Long COVID Assessment: Differentiate cancer-related symptoms (e.g., chemotherapy-induced fatigue) from long COVID using validated tools or biomarkers. Drug Interaction Analysis: Provide detailed pharmacokinetic data on ritonavir interactions with cancer therapies.
3.There are also some areas for improvement in the field of statistics.. Adjust for confounding variables (e.g., vaccination status, tumor stage) using multivariate models. Address wide confidence intervals (e.g., hospitalization rate: 0.5–6.6%) by increasing sample size.
4.I think there are some aspects of the research results that need improvement. Long COVID findings (12.2%) are based on a small subset (15 patients), limiting statistical power. Survival comparisons between reinfected and non-reinfected groups lack significance (HR: 0.61, p=0.32), weakening claims about hybrid immunity benefits. Generalizability is constrained by the Omicron-era focus and exclusion of hematologic malignancies.
- In terms of references Limited discussion of cancer-specific vaccine efficacy (e.g., Wankhede et al. 2023 meta-analysis). No references to guidelines for managing drug interactions in cancer patients (e.g., ASCO or ESMO statements).
- Regarding the tables and figures in the article, Table 1/2 clearly summarize patient demographics and cancer subtypes. However, comorbidities like COPD (36.4%) and diabetes (29.9%) are underdiscussed in the context of COVID-19 outcomes. Missing data on vaccination types (e.g., mRNA vs. viral vector) limits interpretation of immune responses. Kaplan-Meier curves lack stratification by treatment type (e.g., nirmatrelvir vs. remdesivir). Long COVID data (Figure 2) disproportionately focuses on breast cancer patients (86.6%), raising concerns about selection bias.
Author Response
This topic is of some interest and the manuscript is easy to follow.
However, I have the following comments:
-
The manuscript provides real-world data on long COVID outcomes in cancer patients, a poorly understood area and examines drug safety in combination with active cancer therapies, addressing concerns. However, there are still some shortcomings, such as, the single-center, ambispective design and lack of a control group reduce the novelty compared to randomized trials. There are some suggestions for improvement regarding the research methodology Control Group: Include untreated or placebo-controlled patients to isolate the effect of antiviral therapies. Multicenter Collaboration: Expand recruitment to improve generalizability and reduce selection bias. Long COVID Assessment: Differentiate cancer-related symptoms (e.g., chemotherapy-induced fatigue) from long COVID using validated tools or biomarkers. Drug Interaction Analysis: Provide detailed pharmacokinetic data on ritonavir interactions with cancer therapies.
Author response: We fully recognize that the absence of a control group limits our ability to evaluate the specific effects of antiviral therapies in the context of long COVID outcomes in cancer patients. Given the nature of our study, the decision not to include a control group was influenced by ethical consideration. We also agree that a multicenter design would improve the generalizability of the study results and reduce potential selection bias and in future research, we will expand recruitment to multiple centers to enhance the diversity and generalizability of the findings. Due to the limitations of our study design, we were unable to provide comprehensive pharmacokinetic data on drug interactions in the current analysis. However, we recognize that drug-drug interactions can significantly affect treatment efficacy and safety in cancer patients, and more detailed pharmacokinetic studies would be valuable.
There are also some areas for improvement in the field of statistics.. Adjust for confounding variables (e.g., vaccination status, tumor stage) using multivariate models. Address wide confidence intervals (e.g., hospitalization rate: 0.5–6.6%) by increasing sample size.
Author response: We are not able to increase sample size, given the available data and the end of the epidemic. Also, multivariable models could not be fitted given the low number of events; so we fitted a bivariable Cox model to adjust for previous vaccination with no evidence of confounding. As All detahs occurred in Stage 4, we were not able to adjust for this variable. We introduce a statement in the results. Furthermore, the rate of early re-hospitalization being very low (3 patients), its very wide 95%CI points to the lack of precision of this estimate
“This association was not confounded by previous vaccination (HR 0.59, 95%CI 0.22-1.58).”
We recognized these limits in our manuscript and we have now extended this paragraph
“It is a single-center cohort study and thus with a limited sample size of 131, that does not allow for narrow confidence intervals. However, this one center study would guarantee a higher homogeneity of behaviors. We do not have a control arm, which would have been unethical at the time.”
I think there are some aspects of the research results that need improvement. Long COVID findings (12.2%) are based on a small subset (15 patients), limiting statistical power. Survival comparisons between reinfected and non-reinfected groups lack significance (HR: 0.61, p=0.32), weakening claims about hybrid immunity benefits. Generalizability is constrained by the Omicron-era focus and exclusion of hematologic malignancies.
Author response: The HR of 0.61 refers to previous infection and is answered above.
In terms of references Limited discussion of cancer-specific vaccine efficacy (e.g., Wankhede et al. 2023 meta-analysis). No references to guidelines for managing drug interactions in cancer patients (e.g., ASCO or ESMO statements).
Author response: We improved the list of references
Regarding the tables and figures in the article, Table 1/2 clearly summarize patient demographics and cancer subtypes. However, comorbidities like COPD (36.4%) and diabetes (29.9%) are underdiscussed in the context of COVID-19 outcomes. Missing data on vaccination types (e.g., mRNA vs. viral vector) limits interpretation of immune responses. Kaplan-Meier curves lack stratification by treatment type (e.g., nirmatrelvir vs. remdesivir). Long COVID data (Figure 2) disproportionately focuses on breast cancer patients (86.6%), raising concerns about selection bias.
Author response: In Italy, all cancer patients received mRNA vaccine. As I pointed out in the discussion long COVID is more frequent in the female sex and this may justify the detection of long COVID in the population with breast cancer.Of course the small sample size does not allow generalization of the results.
Reviewer 4 Report
Comments and Suggestions for Authors
Review of the paper entitled “Real-world experience with the available outpatient COvid-19 THErapies in patients with canceR (CO.THER)” by Angioletta Lasagna, Giulia Gambini, Catherine Klersy, Simone Figini, Sofia Marino, Paolo Sacchi and Paolo Pedrazzoli
My comments
The topic taken up by the authors is interesting and very important for practical reasons.
There is no clear position of specialists on the impact of SARS-Cov-2 infection on the course of cancer. On the one hand, weakening the immune system is a factor increasing the risk of severe infection. On the other hand, however, disabling the immune response as a result of the disease, and especially as a result of treatment, may be a factor protecting against the most severe forms of COVID-19.
I would like the authors to briefly discuss this issue in their paper.
Some literature data, including meta-analyses, confirm the unfavorable impact of active SARS-CoV-2 infection on the postoperative course of surgical treatment in patients with malignant tumors.
I also ask the authors to briefly discuss this problem.
Author Response
There is no clear position of specialists on the impact of SARS-Cov-2 infection on the course of cancer. On the one hand, weakening the immune system is a factor increasing the risk of severe infection. On the other hand, however, disabling the immune response as a result of the disease, and especially as a result of treatment, may be a factor protecting against the most severe forms of COVID-19. I would like the authors to briefly discuss this issue in their paper.
Author response: We appreciate the suggestion to address this issue in the manuscript and agree that the interaction between cancer, immune function and COVID-19 outcomes is an important and nuanced topic. Unfortunately, given the nature of the study and the endpoints we had set for ourselves, it is not possible to extrapolate immunological analyses
Some literature data, including meta-analyses, confirm the unfavorable impact of active SARS-CoV-2 infection on the postoperative course of surgical treatment in patients with malignant tumors.
I also ask the authors to briefly discuss this problem.
Author response: Thank you for your insightful comment. We agree that the impact of active SARS-CoV-2 infection on the postoperative course of cancer surgery is an important topic, but this issue is outside the scope of our current study. Our research primarily focuses on the role of early anti viral therapies in cancer patients, rather than the specific effects of active COVID-19 infection during the perioperative period. Since our study does not involve patients undergoing surgery or examine the immediate postoperative course, we feel that addressing this particular issue would not be directly relevant to the objectives and scope of our paper. However, we acknowledge the significance of this concern and recognize that future research exploring the intersection of COVID-19 infection and cancer surgery could further elucidate these critical aspects.
Reviewer 5 Report
Comments and Suggestions for Authors
- Grammar and spell check is required.
- Correct the spelling error in Kaplan Meier.
- Introduction part need to be improved with more statements on cancer, COVID19 and its related pictorial representation.
- What type of cancer analyzed in this study? Include that in the data collection section.
- What is the correlation of choosing diabetes, CVS patients in this study?
Grammar and spell check is required.
Author Response
Grammar and spell check is required.
Author response: English has been revised in order to correct stylistic errors
Correct the spelling error in Kaplan Meier.
Author response: Done, Thank you
Introduction part need to be improved with more statements on cancer, COVID19 and its related pictorial representation.
Author response: We have expanded the introduction section
What type of cancer analyzed in this study? Include that in the data collection section.
Author response: Table 2 shows the tumour types considered
What is the correlation of choosing diabetes, CVS patients in this study?
Author response: The decision to include individuals with diabetes and CVD was based on well-established evidence that these conditions are associated with increased susceptibility to severe outcomes in COVID-19 infection.
Reviewer 6 Report
Comments and Suggestions for Authors
The authors of Real-world experience with the available outpatient COvid-19 THErapies in patients with canceR (CO.THER) paper aimed to address the knowledge gap, focusing on the safety of these therapies in combination with the anticancer therapies, and the clinical outcome of COvid-19 THErapies patients.
This paper reported the conclusive data of all the 131 enrolled patients
The authors collected clinical and oncological data from the hospital’s electronic patient 87 records.
Analyses were performed using the Stata software (release 18, StataCorp, College 111 Station, TX, USA).
The dates are presented in 2 tables:
Table 1: general clinical patients’ characteristics
Table 2: oncological patients’ characteristics
The study focused on a Primary endpoint: hospitalization within 14 days and on a Secondary endpoints: SARS-CoV-2 infection and long COVID.
Line 257 – 258: We were not able to perform daily nasopharyngeal swabs to assess promptly negative test results
- Please comment on the above sentence.
Line 261: Conclusions These real-life data can help us understand how to focus all our efforts on cancer patients. However, more data are needed to manage COVID-19 in this frail population better.
- Too much superficial from my point of view. The conclusions should state clearly what the study found.
Lines 42 – 55: Introduction section should also be
Author Response
Line 257 – 258: We were not able to perform daily nasopharyngeal swabs to assess promptly negative test results
Please comment on the above sentence.
Author response: We were unable to perform daily nasopharyngeal swabs to assess negative test results due to logistical and resource-based challenges. Our study design was not aimed at assessing viral shedding. While viral load and time to negative test results are important, the primary objective of our study was to evaluate the efficacy of antiviral therapies. Daily testing was not essential to achieving this goal.
Line 261: Conclusions These real-life data can help us understand how to focus all our efforts on cancer patients. However, more data are needed to manage COVID-19 in this frail population better.
Too much superficial from my point of view. The conclusions should state clearly what the study found.
Author response: We improved the discussion and the conclusion sections.
Lines 42 – 55: Introduction section should also be discussed
Author response: We have expanded the introduction section
Reviewer 7 Report
Comments and Suggestions for Authors
Dear authors,
the manuscript is interesting however it must be improved. My recommendations for improvement are:
- Please extend the introduction, add more references. Explicitly define the novelty of your research and define the research hypotheses.
- Please add grids to all plots in manuscript to improve the readability.
- The conclusions should be extended it is too short. The conclusion should contain one paragraph about what was done in the paper. One paragraph providing the answer to hypotheses defined in the Introduction section based on the discussion section. Provide advantages and disadvantages of proposed research methodology and if possible add some directions for the future work.
Author Response
Please extend the introduction, add more references. Explicitly define the novelty of your research and define the research hypotheses.
Author response: We have expanded the introduction section
Please add grids to all plots in manuscript to improve the readability.
Author response: We improved the readability
The conclusions should be extended it is too short. The conclusion should contain one paragraph about what was done in the paper. One paragraph providing the answer to hypotheses defined in the Introduction section based on the discussion section. Provide advantages and disadvantages of proposed research methodology and if possible add some directions for the future work.
Author response: We improved all sections of the text according to your suggestions
Round 2
Reviewer 3 Report
Comments and Suggestions for Authors
None